# Effects of Arbuscular Mycorrhizal Fungi on *Robinia pseudoacacia* L. Growing on Soils Contaminated with Heavy Metals

**DOI:** 10.3390/jof9060684

**Published:** 2023-06-19

**Authors:** Liuhui Zhao, Tao Yang, Jinxing Zhou, Xiawei Peng

**Affiliations:** 1School of Civil Engineering, Southwest Jiaotong University, Chengdu 610031, China; 2021340107@my.swjtu.edu.cn; 2China Railway First Survey & Design Institute Group Co., Ltd., Xi’an 710043, China; 3College of Biological Sciences and Technology, Beijing Forestry University, Beijing 100083, China; yangtao9798@bjfu.edu.cn; 4School of Soil and Water Conservation, Beijing Forestry University, Beijing 100083, China; zjx001@bjfu.edu.cn

**Keywords:** lead/zinc contamination, *Glomus mosseae*, *Glomus intraradices*, phytoremediation, growth substrates

## Abstract

Arbuscular mycorrhizal fungi (AMF) have been shown to assist plants in increasing metal tolerance and accumulation in heavy metal (HM)-contaminated soils. Herein, a greenhouse pot experiment was conducted to assess the interactions of growth substrates (S1, S2, and S3, respectively) with various HM contamination and nutrient status sampling from a typical contaminated soil and tailings in Shuikoushan lead/zinc mining in Hunan province, China, and AMF inoculation obtained from plants in uncontaminated areas (*Glomus mosseae*, *Glomus intraradices,* and uninoculated, respectively) on the biomass and uptake of HMs and phosphorus (P) by the black locust plant (*Robinia pseudoacacia* L.). The results indicated that the inoculation with AMF significantly enhanced the mycorrhizal colonization of plant roots compared with the uninoculated treatments, and the colonization rates were found to be higher in S1 and S2 compared with S3, which were characterized with a higher nutrient availability and lead concentration. The biomass and heights of *R. pseudoacacia* were significantly increased by AMF inoculation in S1 and S2. Furthermore, AMF significantly increased the HM concentrations of the roots in S1 and S2 but decreased the HM concentrations in S3. Shoot HM concentrations varied in response to different AMF species and substrate types. Mycorrhizal colonization was found to be highly correlated with plant P concentrations and biomass in S1 and S2, but not in S3. Moreover, plant biomass was also significantly correlated with plant P concentrations in S1 and S2. Overall, these findings demonstrate the interactions of AMF inoculation and growth substrates on the phytoremediation potential of *R. pseudoacacia* and highlights the need to select optimal AMF isolates for their use in specific substrates for the remediation of HM-contaminated soil.

## 1. Introduction

Heavy metals (HMs) have little mobility in soil, and are not easily leached via water or degraded by microorganisms [1]. Soil pollution by HMs is attracting increasing attention due to the persistence and hazardous effects of HMs on plant growth, such as the disruption of metabolic processes, nutrient homeostasis, and beneficial microbial activities. HMs are also transferred up the food chain via crops and therefore pose a great threat to human health [2]. HM pollution mainly originates from automobile emissions, industrial waste, and the continuous exploitation and refining of mineral resources [3,4]. Among these sources, metal mining and smelting activities are the most impactful. Although the exploitation and utilization of mineral resources promote China’s economic development, HM contamination is still attracting attention. Therefore, there is an urgent need for the remediation of HM-contaminated soil caused by the metal mining and smelting activities.

Various chemical and physical remediation methods have been used in the restoration of HM-contaminated soils, although their high cost has seriously restricted their general utilization [5]. Phytoremediation is a promising method for the clean-up of HM-contaminated soils as it can be performed in situ, inexpensively, and effectively via the employment of hyperaccumulators that are extremely tolerant to HMs present in the soil environment [6,7,8]. HM-hyperaccumulating plant species (such as *Cannabis sativa*, *Sedum alfredii*, and *Pteris vittata*) can take up and store elevated concentrations of HMs without suffering from metal toxicity or cell damage [9,10,11]. Higher growth rates can lead to the production of large amounts of biomass, thus ensuring efficient phytoremediation [12]. However, most hyperaccumulators grow slowly under high HM concentrations, and thus the phytoextraction process of HMs is limited by the biomass [13]. Moreover, the viability of hyperaccumulators in different polluted environments has become a major issue in phytoremediation [14,15].

The efficiency of phytoextraction primarily depends on the translocation efficiency of HMs from the soil to the plants as well as the plant biomass. Arbuscular mycorrhizal fungi (AMF), which establish a symbiotic interaction with 90% of terrestrial plant species, are involved in the transport of HMs from the soil to the plants by establishing a direct link between the soil and the roots of the host plant [16]. The formation of arbuscular mycorrhizas in association with the AMF have shown that they can reduce HM toxicity in plants and promote plant growth. For example, the enhanced mycorrhizal colonization of Alfred stonecrop (*Sedum alfredii Hance*) with both *Glomus caledonium* and *Glomus mosseae* was found to be able to reduce the translocation of the HMs to the shoots by binding the HMs to the cell walls of the fungal hyphae [10]. Therefore, AMF can act as a filtration barrier against the transfer of the HMs to the plant shoots, which is critical to alleviate HM toxicity [17]. In addition, mycorrhizal association has also been found to improve nutrient uptake—particularly phosphorus (P), a macronutrient that often limits primary productivity in terrestrial ecosystems—indirectly by optimizing phytoremediation via promoting plant growth [18,19,20]. 

Recently, the inoculation of AMF in hyperaccumulators has been suggested to enhance the efficiency of phytoremediation. For instance, inoculation with *Funneliformis mosseae* significantly improved *Solanum nigrum* L. growth and phytoremediation efficiency in cadmium (Cd)-contaminated soil [21]. Black locust (*Robinia pseudoacacia* L.), frequently found in HM-polluted areas and commonly colonized by AMF, is a promising candidate for phytoremediation [22]. Inoculation with AMF can immobilize lead (Pb) in the roots and stems of *R. pseudoacacia* and alleviate the toxic effects of Pb on root development in Pb-contaminated soils [23]. Moreover, elevated temperatures and carbon dioxide were found to promote the removal of HMs by *R. pseudoacacia* seedling roots associated with AMF in Cd-contaminated soils [24,25]. These studies showed that the symbiotic functions of AMF in the remediation of HM-polluted soils are greatly influenced by the environmental conditions. However, AMF-assisted phytoremediation in HM-contaminated soils varies according to the AMF species and growth substrate, and is dependent on plant–fungus–soil combinations [26,27,28]. HMs inhibit AMF spore germination, hyphal extension, and colonization, and the inhibition can be intensified with increasing HM levels and is dependent on the HM species [29]. It has been repeatedly observed that most soil ecosystems in abandoned mining areas are polluted with multiple HM species and have varying physical and chemical properties [30,31]. Following inoculation with AMF, various effects have been observed in phytoremediation. For example, maize (*Zea mays* L.) has demonstrated a great variation in HM uptake in response to AMF colonization on different growth substrates; *Funneliformis mosseae* was found to be the most effective in maize development on recently discharged coal mine spoils and may be the most appropriate for the revegetation of this substrate, while *Rhizophagus intraradices* was determined to be the most beneficial in weathered and spontaneously combusted coal mine spoils [26]. To guarantee the feasibility of AMF-assisted phytoremediation, it is important to determine how AMF inoculation reacts to HM stress, and to explore the optimal AMF–plant–soil combinations in order to improve their efficiency. 

The aims of this study were to assess the effects of AMF on plant growth and HM uptake and accumulation using *R. pseudoacacia* grown in three substrates contaminated with different HM levels, and to explore the relationship between mycorrhizal colonization, plant biomass, and P and HM uptake by *R. pseudoacacia*. Here, we hypothesized that the impacts of AMF inoculation on the phytoremediation potential may be mediated by the interactions between the AMF species and the growth substrates. This study will provide experimental evidence for use of the pioneer species *R. pseudoacacia* in a symbiotic association with the appropriate AMF to remediate HM-contaminated soils.

## 2. Materials and Methods

### 2.1. Growth Substrate Preparation

The survey was conducted in the Shuikoushan Pb/Zn mining and smelting area in Hunan province, China, located between the latitudes 26°25′58″–26°36′11″ and longitudes 112°22′34″–112°42′58″, respectively. The mine was created in 1896 and is still one of the largest Pb and Zn production bases in China. It generates a significant quantity of dust and tailings every year, and contaminates the surrounding areas over several kilometers [32]. According to the pollution situation around the mine and contamination degree, the topsoil (0–20 cm) of three types of sampling sites was selected for this study, including soil contaminated by the collapse of the tailings dam (S1); soil in the upper layers of the tailings pond (S2); and the downwind area around the smelter which was contaminated by the serious smoke and dust pollution (S3). Three plots were selected in each of these fields, and 5 random topsoil samples were collected and then homogenized to a composite sample at each plot after the removal of stones. The soil was air-dried and sieved through a 2 mm sieve for analysis of the physical and chemical properties. The organic matter (OM) content was determined using the potassium dichromate heating method [33]. Available nitrogen (AN) contents of air-dried soils were extracted in 50 mL 0.5 M K_2_SO_4_ at a ratio of 1:4 (*w*/*v*) and determined on an Elementar analyzer using a dry combustion method (Vario MAX CN, Germany; detection limit = 0.1%). The contents of dissolved available P were extracted in NaHCO_3_ at a ratio of 1:20 (*w*/*v*), and were determined according to the molybdenum antimony anti-colorimetric method [34]. The HM concentrations in the topsoil were determined using inductively coupled plasma-optical emission spectroscopy (Optima 7000 DV, PerkinElmer, Waltham, MA, USA) following HNO_3_^−^-HCl-HClO_4_ digestion. The basic physical and chemical properties of the substrates are shown in Table 1. Prior to the pot experiment, the three substrates were autoclave-sterilized at 121 °C for 2 h to eliminate the indigenous AMF propagules and other microorganisms. 

### 2.2. Plant and AMF Preparation

Seeds of *R. pseudoacacia* collected from the Shuikoushan Pd/Zn mine aera were supplied by the Forestry Research Institute of the Chinese Academy of Forestry Sciences (Beijing, China). Before sowing, the seeds were surface-sterilized and pre-germinated on moist filter paper at 25 °C for ~48 h until radicles appeared. The two AMF isolates used in this study were *Glomus mosseae* and *Glomus intraradices*, which were derived from plants located in uncontaminated areas, and supplied by Runjin Liu (Institute of Mycorrhizal Biotechnology, Qingdao Agricultural University, Qingdao, China). The AMF isolates were propagated on maize (*Zea mays* L.) and white clover (*Trifolium repens* L.) in a soil–sand mixture (1:1 *w/w*) for 4 months. The resulting inoculum consisted of cultivation substrate-containing spores (approximately 15 spores per gram), extraradical mycelium, colonized maize, and white clover root fragments. 

### 2.3. Pot Experiment

The pot experiment was conducted with a 3 × 3 factorial combination, which comprised of three substrates (S1, S2, and S3) and three AMF inoculations (*G. mosseae* inoculation, *G. intraradices* inoculation, and no inoculation). The total weight of the substrates (S1, S2, and S3) of 500 g, 650 g, and 830 g, respectively, were transferred into per plastic pots (top diameter 20 cm, bottom diameter 15 cm, and height 15 cm, respectively) to provide a depth of 8 cm. Three pots were used for each treatment, forming a total of 27 pots, and 10 black locust seeds were sown into each pot. Then, separately, 250 g, 325 g and 415 g substrates of S1, S2, and S3 were covered accordingly, and a total depth of growth substrate of 12 cm per pot for S1, S2, and S3, respectively, was formed as a result. At the same time, inoculation treatment was conducted by mixing 45 g inoculum into each pot 4 cm below the soil surface. Sterilized inoculum (autoclaved at 121 °C for 2 h) were used as the uninoculated treatments. Then, plants were grown in a glasshouse with temperature control (25 °C day/15 °C night), and the soil moisture was maintained at ~60% of the water-holding capacity by adding sterile water during the experimental period. The seedlings in each pot were thinned to five after 1 week.

### 2.4. Sampling and Analysis

After 12 weeks, all plants from each pot were harvested and divided into the shoots and roots. The roots were carefully washed with sterile water to remove the adhering soil and sand particles. Fibrous root fragments were cut into 1 cm pieces, stained with 0.05% trypan blue in lactoglycerol, and mounted on glass slides (20 fragments per slide) for the mycorrhizal colonization examination according to the gridline intersect method after clearing with 10% KOH and staining with acid fuchsin [35,36]. For each sample, 120 root fragments were examined to estimate the mycorrhizal colonization rate. The biomass of the shoots and roots was determined after oven drying at 70 °C for 48 h. Root and shoot P concentrations were measured using molybdenum antimony colorimetric methods after digestion using a microwave digestion system (μPREP-A, MLS, Leutkirch, Germany). Oven-dried subsamples were ground, and 0.3–0.5 g of the samples were digested using 5 mL high-purity HNO_3_ at 120 °C in an open block digestion system (AIM600, Aim Lab Pty Ltd., Queensland, Australia). The acid digests were then diluted with ultra-pure water and brought up to 50 mL. The concentrations of Cd, Zn, copper (Cu), arsenic (As), and Pb were determined using inductively coupled plasma-optical emission spectroscopy (Optima 7000 DV, PerkinElmer, USA). The HM accumulation in plants was calculated based on Equation (1), according to Wei et al. [37].
HM accumulation = dry biomass × HM concentration in tissue(1)

### 2.5. Statistical Analyzes

All datasets were tested for normality and homogeneity before analysis. One-way analysis of variance with the Tukey’s HSD post-hoc test was utilized to identify significant differences in mycorrhizal colonization, plant dry weight and height, and the accumulation and concentrations of P and HM among the growth substrates (*p* < 0.05). Furthermore, the difference between the incubation treatments within the same growth substrate were evaluated using the one-way analysis of variance and Tukey’s HSD post-hoc test (*p* < 0.05). Simple linear regression models were used to analyze the relationships of mycorrhizal colonization with the biomass and height of *R. pseudoacacia* and the plant P concentration. To explore the beneficial role of P in plant growth, regression analysis was performed to assess the relationships between the plant P concentration and the biomass. All analyzes and plots were performed using R software v3.4.2 accessed on 7 October 2017 (https://www.r-project.org/). 

## 3. Results

### 3.1. Soil Characteristics

The main characteristics of the soil samples from each substrate are shown in Table 1. The organic matter (OM) and available N (AN) content were found to be much lower in S1 than in the other soil samples (Table 1, *p* < 0.05). However, the available P (AP) content was much lower in S2 (Table 1, *p* < 0.05). The AN and AP contents were highest in S3 (Table 1, *p* < 0.05). Soil pH ranged from 7.75 to 7.87, respectively. The concentrations of Cd, Pb, and Zn in the three soil samples were found to be far higher than the acceptable pollution levels set in the Environmental Quality Standards for Soils. However, we only found excessive Cu contamination in the S2 and S3 soil samples, and excessive As contamination in S2 alone. The concentrations of Zn, Cu, As, and Cd were found to be the highest in S2, whereas that of Pb was the highest in S3 (Table 1, *p* < 0.05). 

### 3.2. Mycorrhizal Colonization and Plant Dry Weight

The effects of AMF inoculation on the plant dry weight and mycorrhizal root colonization are summarized in Table 2. In the three soil substrates, inoculation with AMF induced different responses in the mycorrhizal colonization of *R. pseudoacacia* (Table 2, *p* < 0.05). In S1, the roots were all highly colonized by the two AMF species, and there was no significant differences observed among the inoculation treatments. In S2, the colonization rate of *G. intraradices* was found to be significantly higher than that of *G. mosseae* (Table 2, *p* < 0.05), while the opposite was observed in S3. The rates of mycorrhizal colonization were significantly increased by 259% (S1) and 328% (S2) with *G. intraradices*-inoculated treatment, and by 169% (S1) and 198% (S2) with *G. mosseae*-inoculated treatment, compared with S3 (Table 2), respectively. 

Plant height and shoot dry weight in S3 was found to be much greater than that in S1 and S2, while assessment of the root dry weight showed a opposite trend (Table 2, *p* < 0.05). Inoculation with *G. mosseae* or *G. intraradices* significantly increased the dry weights of the roots and shoots in S1 and S2 compared with the uninoculated treatments. In S2, inoculation with *G. intraradices* increased the plant dry weight more significantly compared with *G. mosseae* inoculation (Table 2, *p* < 0.05). However, inoculation with the two AMF had no significant effects on the plant height or dry weight in the S3 soil sample (Table 2, *p* < 0.05). We also calculated the relationship between mycorrhizal colonization and the plant growth (Figure 1). The shoot, root dry weights, and heights of *R. pseudoacacia* were found to be positively related to mycorrhizal colonization in the S1 and S2 soil samples (Figure 1, *p* < 0.01). However, the plant dry weight and height in the S3 soil sample was determined to be poorly correlated with mycorrhizal colonization (Figure 1, *p* > 0.05).

### 3.3. Plant Uptake of Available P

*Robinia pseudoacacia* L. planted on the S3 soil sample was found to greatly enhance the accumulation and concentration of P compared to the S1 and S2 samples (Figure 2, *p* < 0.05). Twelve weeks after inoculation with *G. mosseae* or *G. intraradices*, there were significant elevations in P concentration and accumulation in the shoots and roots observed, which were more pronounced with *G. intraradices* inoculation (Figure 2, *p* < 0.05), in S1 and S2; in S1, P concentrations in the shoots and roots showed more significant improvements. However, in S3, application of the AMF decreased the P concentration and accumulation in the shoots (Figure 2a,c, *p* < 0.05), but had no significant effects on the P concentration in the roots (Figure 2b, *p* < 0.05). In S1 and S2, the root and shoot P concentrations increased in correlation with the increases observed in the mycorrhizal colonization rate, whereas no significant correlation was found in S3 (Figure 3, *p* < 0.05). Furthermore, the dry weights of the roots and shoots only exhibited significant positive relationships with the plant P concentrations in the S1 and S2 soil samples (Figure 4, *p* < 0.05).

### 3.4. Uptake and Accumulation of Heavy Metals 

The concentrations of all HMs present in the roots showed a significant difference in response to the growth substrates, while in the shoots AS concentrations were found to not be influenced by the growth substrates (Figure 5 and Figure 6, *p* < 0.05). Inoculation with *G. intraradices* or *G. mosseae* was found to significantly increase the concentrations of all HMs in the roots of *R. pseudoacacia* in S1, with no significant difference observed between the two AMF species (Figure 5, *p* < 0.05). Except for Cu-ions, the application of AMF was found to enhance the uptake of all HMs in S2, and the concentrations of Zn, Pb, and As were found to be higher in the roots inoculated with *G. mosseae* than in roots inoculated with *G. intraradices* (Figure 5b, *p* < 0.05). AMF inoculation decreased the concentrations of all HMs in the roots in the S3 soil sample (Figure 5, *p* < 0.05). Meanwhile, the concentrations of all HMs in the shoots showed more complex trends; inoculation with the two AMF species significantly elevated the concentrations of Cd and As in S1, but reduced them in S2 (Figure 6a,d, *p* < 0.05). In S3, inoculation with *G. mosseae* significantly decreased the Cd concentration, but increased the As concentration in *R. pseudoacacia* shoots (Figure 6a,e, *p* < 0.05). Across all substrates, inoculation with *G. mosseae* had no effect on the shoot Cu concentration, whereas inoculation with *G. intraradices* increased the shoot Cu concentration in the S1 and S2 soil samples (Figure 6b, *p* < 0.05). In the shoots, AMF inoculation was found to exhibit no effects on the Zn concentration (Figure 6c, *p* < 0.05), whereas it was determined to significantly decrease the Pb concentration in S1 and increase the Pb concentration in S2, respectively (Figure 6d, *p* < 0.05). 

The accumulation of all HMs in *R. pseudoacacia* roots were found to be significantly higher in S1 and S2 than S3, except for Cd ions (Table 3, *p* < 0.05). In S1, inoculation with AMF significantly increased the accumulation of all HMs in *R. pseudoacacia* shoots and roots, but there was no significant difference observed between *G. intraradices* and *G. mosseae* (Table 3 and Table 4, *p* < 0.05). Similarly, the HM accumulation in plants inoculated with *G. intraradices* or *G. mosseae* was found to be significantly higher than in the uninoculated treatments in S2, particularly in the treatments inoculated with *G. intraradices* (Table 3 and Table 4, *p* < 0.05). However, inoculation with AMF significantly decreased the root HM accumulation in S3 (Table 3, *p* < 0.05). In S3, shoot Cd and Pb accumulation were increased with *G. intraradices* treatment (Table 4, *p* < 0.05) but decreased with *G. mosseae* treatment. However, the application of *G. intraradices* or *G. mosseae* had no effects on shoot Zn or Cu accumulation in the S3 soil sample but increased As accumulation (Table 4, *p* < 0.05). 

## 4. Discussion

Mycorrhizal colonization is an important indicator for evaluating the establishment of a symbiotic relationship between AMF and a plant host [21]. In this study, symbiotic relationships were successfully established to varying degrees between the AMF species and *R. pseudoacacia* grown in three substrates with different pollution, physical, and chemical properties (Table 2). *Glomus intraradices* and *Glomus mosseae* exhibited strong adaptability to HM-polluted environments, even though they were not isolated from metalliferous soils. However, the two AMF strains acted differently in S1, S2, and S3 (Table 2), indicating the differing ecological adaptabilities of these two strains. In comparison with S3, characterized with the highest available P, the colonization rates of both AMF strains were found to be significantly higher in S1 and S2 (Table 2). This phenomenon was deemed to mainly be due to the differences in P concentrations among the substrates, which may have caused varying plant physiological activities, such as rhizodeposition characteristics, which stimulate or suppress the activity of the AMF community and the potential of mycorrhizal colonization [38]. 

Phosphorus is one of the most limiting nutrients for plant growth and terrestrial ecosystem productivity [39]. For example, high P concentrations in mycorrhizal plants can lead to a higher RNA production to meet protein synthesis needs, resulting in improved plant growth rates [26]. The formation of arbuscular mycorrhizas in association with AMF is an essential strategy for host plant P acquisition to support growth. For example, AMF hyphae can release insoluble P, thereby contributing to enhanced P uptake [40,41]. However, the strains evaluated in this study showed no effect on the plant dry weights and height in the S3 soil sample (Figure 1 and Table 2). A previous study demonstrated that a variety of factors may influence the beneficial effects of AMF on plant growth, such as AMF tolerance to HMs, drought, and nutrient limitation [31,42]. In this study, plant dry weight increased with the plant P concentration in S1 and S2 (Figure 4 and Table 2), whereas it did not change in S3. This is possibly because the higher P availability in S3 than in S1 and S2 may eliminate plant P limitation as a result (Table 1), and thereby eliminate the influence of mycorrhizal colonization on plant growth (Figure 1 and Figure 3) [43,44]. On the other hand, the increased Pb content observed in S3 may constrain the colonization rates of both AMF strains (Table 2). Thus, a combination of higher Pb and AP concentrations in S3 might be unfavorable for the application of AMF for phytoremediation.

It has been widely reported that high HM contents in growth substrates can cause a higher plant phytotoxicity, resulting in a limited phytoremediation efficiency [45]. The beneficial effects of AMF on phytoremediation have often been related to the regulation of HM acquisition. Understanding such mechanisms is crucial for the optimal application of AMF during phytoremediation activities. AMF inoculation has been shown to both enhance and reduce HM uptake in plant tissues and, in some cases, have no effect on HM uptake which may depend on HM-contaminant levels [46]. Pb, which is one of the most frequent HMs found in the soil and nonessential for plants [47,48], is relatively stable in the soil [49]. AMF inoculation increased the root Pb concentration in S1 and S2 but decreased it in S3 (Figure 5d). This was probably because AMF alleviated the toxic effects of Pb on root development, improved root biomass in S1 and S2 (Table 2), and further immobilized more Pb in the roots as a result [23]. Compared with S1 and S2, soil Pb concentrations were 7.44 and 4.91 times higher in S3, respectively (Table 1). However, the uptake of Pb by the roots and shoots was significantly lower in S3 (Figure 5d and Figure 6d). This phenomenon may have resulted from an excessive Pb contamination in S3, which could have induced more oxidative stress in the plant and AMF, thus being unfavorable to symbiosis formation, and resulting in a very low phytoremediation efficiency, regardless of inoculation [50]. Notably, considering the seriously negative effects of Pb contamination on the phytoremediation of *R. pseudoacacia*, it is therefore necessary to develop novel strategies, including the selection of AMF or plants which harbor a higher Pb accumulation potential and tolerance performance regarding to excessive Pb ions. In plant shoots, AMF inoculation decreased the concentration of Pb in S1 (Figure 5d). This finding was consistent with Sabra et al. [51], who also observed that the Pb concentration was significantly decreased in the shoots after colonization of plants with *Rhizophagus irregularis* or *Serendipita indica*. However, the Pb concentration was increased in AMF-inoculated shoots in S2. The different responses of the shoot Pb concentration to the substrate type may suggest that the AMF-assisted phytoremediation of Pb is substrate-specific.

Similarly, the application of AMF markedly increased root Zn concentrations in S1 and S2 rather than S3, and both AMF-inoculated and uninoculated plant roots had higher Zn concentrations in S2 than in S1 (Figure 5c). This may be due to the large difference in Zn concentrations between S1 and S2 (Table 1). Zn is easy to transfer during plant metabolism, and can exhibit toxic effects on the plant cells when the concentration exceeds a certain range [52,53]. In contrast to Pb, despite excessive Zn concentrations in S2, the application of AMF markedly increased root Zn concentrations in S2, and both AMF-inoculated and uninoculated plant roots had higher Zn concentrations in S2 than in S1 (Figure 5c). Furthermore, uninoculated roots had higher concentrations of HMs, particularly Zn, compared with inoculated roots in S3 (Figure 5). These AMF species may be more suitable for the phytoremediation of Zn-contaminated soils than for the Pb-contaminated soils. In addition, organic matter (OM), a critical factor in controlling the sorption and sequestration of pollutants, was found to be significantly higher in S3, which could thereby buffer the negative effects of excessive Zn ions on AMF inoculation. However, AMF did not affect the shoot Zn concentration in *R. pseudoacacia* plants (Figure 6c). The retention of Zn by the *R. pseudoacacia* roots may be explained by a self-protection mechanism that prevents excess HMs from entering the stems and leaves, thereby reducing the toxic effects of HMs [54].

Soil with a Cd concentration exceeding 0.5 mg/kg is considered as contaminated with a high phytotoxicity [55]. Although the Cd concentration was increased in the AMF-inoculated roots in S1 and S2, the shoot Cd concentration was significantly increased in S1 but decreased in S2, suggesting that AMF-assisted phytoremediation was more suitable in S2 (Figure 5a and Figure 6a). A previous study demonstrated that antioxidant activities in mycorrhizal plants were increased at lower Cd concentrations but decreased at higher Cd concentrations [21]. Cu, as a component of several enzymes, participates in various physiological metabolic processes, and exhibits an important impact on the growth and development of plants. In the present study, soil Cu concentrations were 319% and 293% higher in S2 and S3 than in S1, respectively (Table 1). However, the application of AMF did not increase the Cu concentration in the roots in S2 or S3. This may be related in that soil Cu contamination will lead to the disordering of the plant metabolic process and interfere with the balance between the ions in plants, which leads to very little effects on the Cu uptake in plant roots in response to the application of AMF [56].

In this experiment, inoculation with AMF played an important role in facilitating HM accumulation by *R. pseudoacacia* grown in S1 and S2, especially in roots. Compared with S1 and S2, the root accumulation of HMs, particularly Pb and Zn, was found to be significantly lower in S3 (Table 3). Although *R. pseudoacacia* grown in S3 had the highest shoot dry weight (Table 2), the shoot HM accumulation was the lowest compared with S1 and S2 (Table 4), suggesting that the difference in substrate greatly influenced the beneficial role of AMF in *R. pseudoacacia* phytoremediation. Therefore, it is important to select appropriate AMF species for phytoremediation in a specific environment. Furthermore, we must explore the role of AMF in relieving HM-induced phytotoxicity under HM stress to take the full advantage of their potential value in phytoremediation.

## 5. Conclusions

In three typical mining soils with different levels of HM pollution (S1, S2, and S3), a symbiotic relationship between plants and AMF was well established. *Glomus mosseae* and *Glomus intraradices* significantly promoted growth in S1 and S2, whereas they had no effect in S3, which was characterized with the highest Pb contamination and P availability. Similarly, the P content and accumulation in the shoots and roots of the host plants in S1 and S2 increased significantly after inoculation with AMF, whereas there was no significant change observed in S3. Additionally, AMF exhibited different influences on HM uptake by *R. pseudoacacia*, which depended on the HM concentrations and P availability. Overall, AMF significantly increased the efficiency of phytoextraction in S1 and S2 but decreased this efficiency in S3. Therefore, further research on the enzymes, proteins, and genes involved in the AMF-assisted improvement of the host plant HM resistance is needed.

## Figures and Tables

**Figure 1 jof-09-00684-f001:**
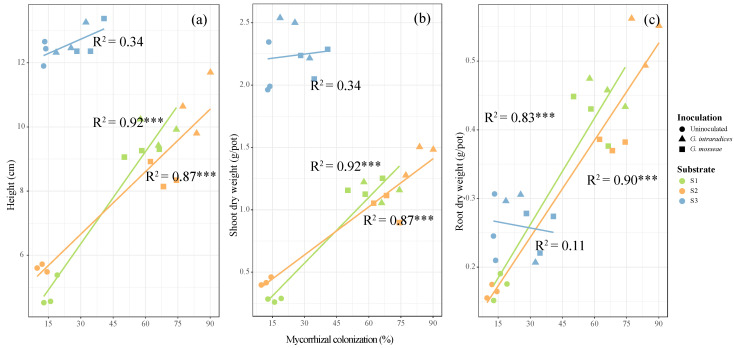
The linear relationships present between the plant height (**a**), shoots (**b**) and roots dry weight (**c**) and mycorrhizal colonization, respectively. Different symbols and color represent different types of substrate and inoculation treatments, respectively. Asterisks indicate the significance of each predictor, with three asterisks indicating *p* < 0.001.

**Figure 2 jof-09-00684-f002:**
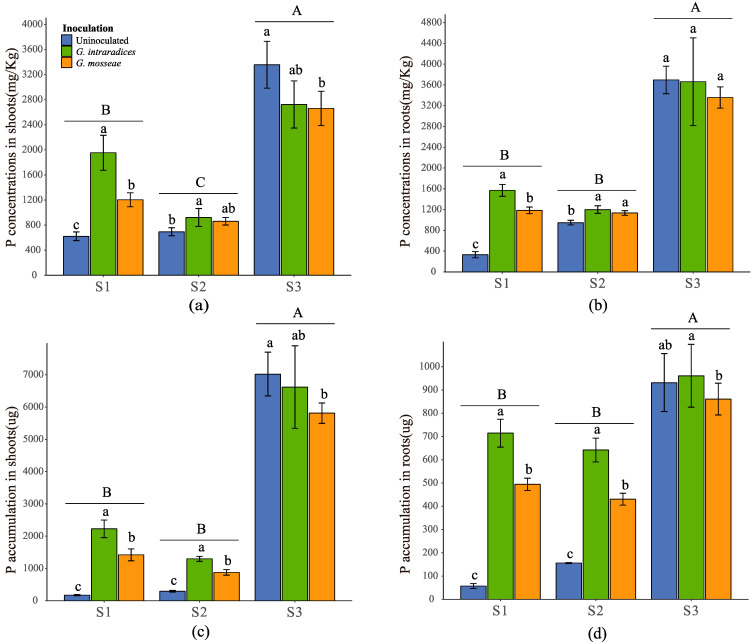
P concentration in the shoots and roots (**a**,**b**), and P accumulation in the shoots and roots (**c**,**d**) of *R. pseudoacacia*, respectively. Different uppercase letters indicate significant differences between the different substrates at *p* < 0.05. Different lowercase letters indicate significant differences between the different AMF inoculations within the same substrates at *p* < 0.05. ns indicates no significant difference among the different AMF inoculations within the same substrates at *p* < 0.05. (P accumulation = dry biomass × P concentration in tissue).

**Figure 3 jof-09-00684-f003:**
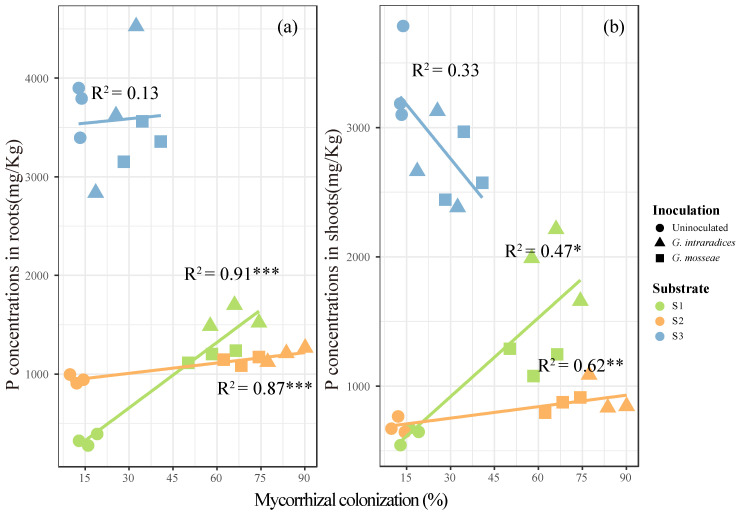
The linear relationships present between the P concentrations of the roots (**a**) and shoots (**b**) and mycorrhizal colonization. Different symbols and color represent different types of substrate and inoculation treatments, respectively. Asterisks indicate the significance of each predictor, with one, two and three asterisks indicating 0.01 *≤ p* < 0.05, 0.001 *≤ p <* 0.01, and *p* < 0.001, respectively.

**Figure 4 jof-09-00684-f004:**
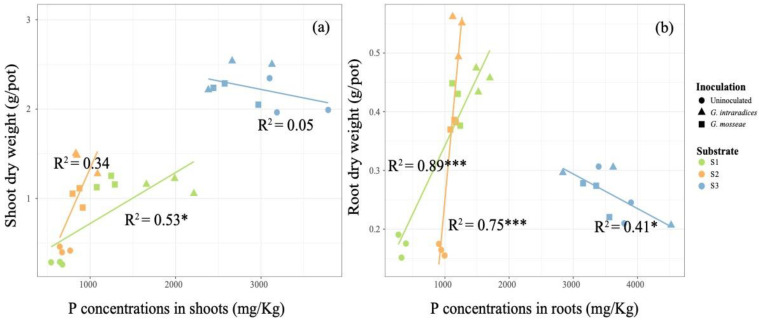
The linear relationships present between the P concentrations of the shoot (**a**) and root (**b**) with plant tissue dry weights, respectively. Different symbols and color represent different types of substrate and inoculation treatments, respectively. Asterisks indicate the significance of each predictor, with one and three asterisks indicating *p* < 0.05 and *p* < 0.001, respectively.

**Figure 5 jof-09-00684-f005:**
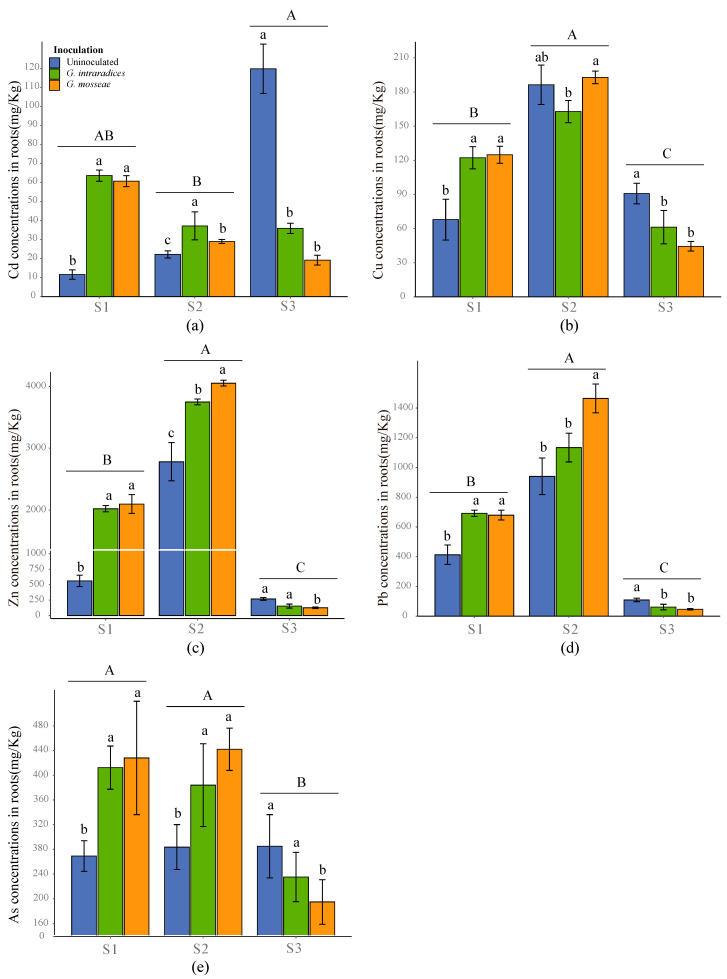
HM concentrations of Cd (**a**), Cu (**b**), Zn (**c**), Pb (**d**) and As (**e**) in *R. pseudoacacia* roots, respectively. Different uppercase letters indicate significant differences between the different substrates at *p* < 0.05. Different lowercase letters indicate significant differences between the different AMF inoculations within the same substrates at *p* < 0.05.

**Figure 6 jof-09-00684-f006:**
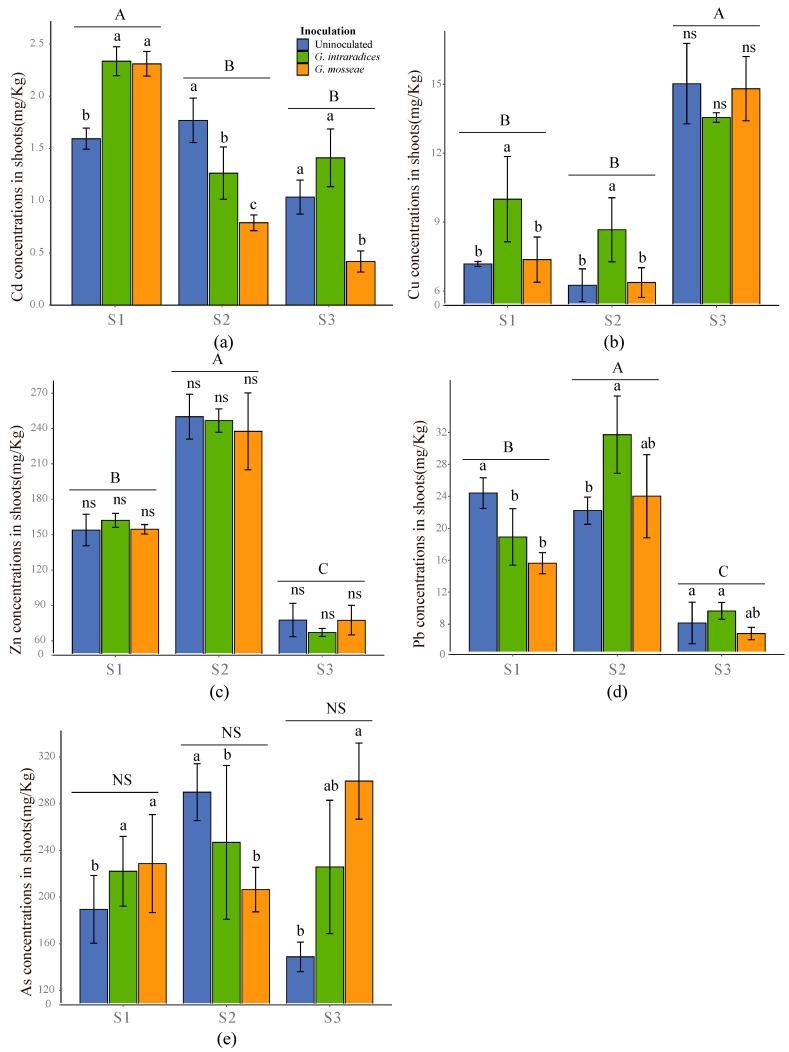
HM concentrations of Cd (**a**), Cu (**b**), Zn (**c**), Pb (**d**) and As (**e**) in *R. pseudoacacia* shoots, respectively. Different uppercase letters indicate significant differences between the different substrates at *p* < 0.05. Different lowercase letters indicate significant differences between the different AMF inoculations within the same substrates at *p* < 0.05. NS indicates no significant difference among the different AMF inoculations within the same substrates at *p* < 0.05.

**Table 1 jof-09-00684-t001:** The characteristics of the three types of substrates.

Parameters	S1	S2	S3
pH	7.87 ± 0.09ns	7.82 ± 0.15ns	7.75 ± 0.23ns
Organic matter (mg/kg)	12.27 ± 2.13c	27.44 ± 3.12a	21.85 ± 2.56b
Available P (mg/kg)	6.0 ± 1.21b	2.5 ± 0.56c	15.0 ± 1.54a
Available N (mg/kg)	10.37 ± 1.89c	16.29 ± 1.34b	24.61 ± 2.12a
Cd (mg/kg)	5.3 ± 0.31c	21.7 ± 2.19a	8.4 ± 1.23b
Zn (mg/kg)	1170.8 ± 78.2c	6619.8 ± 101.2a	2248.7 ± 88.9b
Pb (mg/kg)	549.1 ± 45.1c	831.8 ± 78.1b	4082.2 ± 121.2a
Cu (mg/kg)	58.9 ± 8.2b	187.7 ± 15.2a	173.2 ± 11.1ab
As (mg/kg)	13.2 ± 1.2b	24.7 ± 2.5a	9.6 ± 0.9c

Note: Values in the table are shown as means ± SE (n = 3). Different lowercase letters indicate significant differences among the different substrates at *p* < 0.05. ns indicates no significant difference among the different substrates at *p* < 0.05.

**Table 2 jof-09-00684-t002:** Effect of inoculation with AMF on mycorrhizal colonization and the plant biomass of plants grown in three different substrates.

Substrate	Inoculation	Mycorrhizal Colonization (%)	Height (cm)	Shoot Dry Weight (g/pot)	Root Dry Weight (g/pot)
S1	Uninoculated	16.0 ± 3.1b	4.82 ± 0.49b	0.278 ± 0.015b	0.173 ± 0.020b
	*G. intraradices*	66.0 ± 8.3a	9.86 ± 0.41a	1.145 ± 0.085a	0.455 ± 0.021a
	*G. mosseae*	58.3 ± 8.1aB	9.21 ± 0.13aB	1.178 ± 0.066aB	0.418 ± 0.038aA
S2	Uninoculated	12.1 ± 2.3c	5.60 ± 0.12b	0.425 ± 0.032c	0.165 ± 0.010c
	*G. intraradices*	83.7 ± 6.4a	10.71 ± 0.95a	1.421 ± 0.126a	0.536 ± 0.037a
	*G. mosseae*	68.3 ± 6.0bA	8.47 ± 0.41aB	1.022 ± 0.112bB	0.379 ± 0.009bA
S3	Uninoculated	13.3 ± 0.5c	12.33 ± 0.39	2.100 ± 0.213	0.254 ± 0.049
	*G. intraradices*	25.5 ± 6.9b	12.68 ± 0.51	2.418 ± 0.177	0.270 ± 0.054
	*G. mosseae*	34.5 ± 6.3aC	12.70 ± 0.59A	2.191 ± 0.125A	0.258 ± 0.032B

Note: Values in the table are shown as means ± SE (n = 3). Different uppercase letters indicate significant differences among the different substrates at *p* < 0.05. Different lowercase letters indicate significant differences among the different AMF inoculations within the same substrates at *p* < 0.05.

**Table 3 jof-09-00684-t003:** Effect of AMF inoculation on HM accumulation in the roots of plants grown in three different types of soil.

Substrate	Inoculation	Cd (μg/pot)	Zn (μg/pot)	Pb (μg/pot)	Cu (μg/pot)	As (μg/pot)
S1	Uninoculated	1.99 ± 0.40b	95.70 ± 6.66b	70.92 ± 11.32b	11.71 ± 3.37b	46.56 ± 7.63b
	*G. intraradices*	28.93 ± 1.28a	919.91 ± 43.68a	314.88 ± 5.55a	55.73 ± 6.61a	187.56 ± 15.45a
	*G. mosseae*	25.47 ± 3.40aA	875.83 ± 75.53aB	284.45 ± 28.60aB	52.06 ± 1.85aB	180.12 ± 49.47aA
S2	Uninoculated	3.66 ± 0.40c	458.78 ± 62.79c	155.57 ± 26.21c	30.65 ± 2.23c	46.63 ± 5.09c
	*G. intraradices*	20.05 ± 4.92a	2007.89 ± 147.74a	604.90 ± 21.79a	86.93 ± 1.18a	204.38 ± 25.89a
	*G. mosseae*	11.00 ± 0.60bB	1537.06 ± 17.50bA	554.95 ± 26.64bA	73.13 ± 3.19bA	167.69 ± 14.53bA
S3	Uninoculated	30.23 ± 5.35a	68.28 ± 14.12a	27.72 ± 6.53a	22.81 ± 2.75a	71.16 ± 8.83a
	*G. intraradices*	9.59 ± 1.40b	40.25 ± 2.36b	15.91 ± 2.24b	16.12 ± 3.25b	62.23 ± 8.65ab
	*G. mosseae*	4.94 ± 1.00bC	32.27 ± 3.28bC	11.97 ± 2.31bC	11.52 ± 2.34bC	49.89 ± 9.49bB

Note: Values in the table are shown as means ± SE (n = 3). Different uppercase letters indicate significant differences between the different substrates at *p* < 0.05. Different lowercase letters indicate significant differences between the different AMF inoculations within the same substrates at *p* < 0.05.

**Table 4 jof-09-00684-t004:** Effect of AMF inoculation on HM accumulation in the shoots of plants grown in three different types of soil.

Substrate	Inoculation	Cd (μg/pot)	Zn (μg/pot)	Pb (μg/pot)	Cu (μg/pot)	As (μg/pot)
S1	Uninoculated	0.44 ± 0.05b	42.71 ± 1.29b	6.92 ± 0.68b	2.00 ± 0.08b	52.48 ± 5.59b
	*G. intraradices*	2.67 ± 0.21a	185.93 ± 19.90a	22.18 ± 4.41a	11.48 ± 2.61a	252.70 ± 16.06a
	*G. mosseae*	2.72 ± 0.25aA	182.14 ± 6.89aB	18.99 ± 2.30aB	8.68 1.19aB	268.68 ± 42.93aB
S2	Uninoculated	0.75 ± 0.14b	106.32 ± 13.02c	9.64 ± 1.20c	2.67 ± 0.51c	122.86 ± 8.78c
	*G. intraradices*	1.80 ± 0.42a	351.25 ± 42.46a	45.71 ± 8.17a	12.20 ± 0.80a	352.63 ± 109.44a
	*G. mosseae*	0.81 ± 0.16bB	240.37 ± 5.64bA	24.78 ± 4.64bA	6.50 ± 0.94bB	211.27 ± 34.17bB
S3	Uninoculated	2.15 ± 0.18b	160.99 ± 14.81	17.88 ± 4.57b	31.31 ± 1.16	312.49 ± 40.42b
	*G. intraradices*	3.38 ± 0.49a	161.60 ± 4.78	24.46 ± 3.49a	32.74 ± 1.93	540.19 ± 99.39a
	*G. mosseae*	0.92 ± 0.25cAB	169.24 ± 24.63C	16.03 ± 2.58bB	32.40 ± 2.96A	653.53 ± 31.99aA

Note: Values in the table are shown as means ± SE (n = 3). Different uppercase letters indicate significant differences between the different substrates at *p* < 0.05. Different lowercase letters indicate significant differences between the different AMF inoculations within the same substrates at *p* < 0.05.

## Data Availability

Data will be made available on request.

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
