# Peer review of "Effects of Arbuscular Mycorrhizal Fungi on Robinia pseudoacacia L. Growing on Soils Contaminated with Heavy Metals"

_jof, 2023, doi:10.3390/jof9060684_

Round 1
Reviewer 1 Report
The work is interesting and its results indicate that mycorrhizal fungi can act as a biofilter. These fungi are an effective barrier to the movement of heavy metals from the soil to plants. These results confirm the possibility of using mycorrhization in areas heavily degraded by industry. As for the methodology, I have no comments. I just have a question about the obtained cultures of mycorrhizal fungi Glomus mosseae and Glomus intraradices, which were used in the research. Were they derived from plants grown in contaminated areas or were they pure cultures obtained from plants in uncontaminated areas? According to the research of some authors, strains of fungi should be used for artificial mycorrhization, taking into account not only their mycorrhizal abilities, but also their origin and adaptation to local conditions. This remark may be a clue for the authors in the case of continuation of the research.
Author Response
On behalf of my co-authors, we would like to express our appreciation to you for taking the time to review our paper. We are glad to hear that you recognize the scientific significance of our research. In this research, these AM fungi obtained from plants in uncontaminated areas which donated by professor Runjin Liu from Qingdao Agricultural University. It have been show greatly potential in improving photosynthesis and growth under salt stress and drought conditions. We greatly appreciate your insightful suggestion about considering the origin and adaptation of AMF. Indeed, we are isolating strains from contaminated areas and would fully consider your comments in our future work, which would be helpful for better use and development of the artificial mycorrhization in areas heavily degraded by industry, especially for heavy metals contaminated soils.
Reviewer 2 Report
The text contains a large number of abbreviations of the names of variants, elements, etc., although these abbreviations are given in Chapter 2. In this large text, the abbreviations sometimes cause difficulties in reading. Sometimes it is more advantageous for the recipient to give the full name or memory. All detailed Reviewer's comments and suggestions are marked as comments in yellow colour in the manuscript.

Author Response
On behalf of my co-authors, we would like to express our great appreciation to you for taking the time to review our paper and given these suggestions to improve our manuscript. We are so apologize for our irregular writing and unclear description in our manuscript. We have thoroughly reviewed your suggestions and have made substantial changes in revised manuscript based on your valuable comments. We are confident that our revisions and responses will meet your expectations and hope that our work will be considered for publication in Journal of Fungi. Please see the attachment .

Reviewer 3 Report
1. The background sentence (lines 12-14) did not indicate the need for the study. Therefore, it should be rewritten.
2. The common name of the plant should be used, instead of only the scientific name.
3. In line 15, growth substances have been used equally to the AMF, but the title and keywords did not mention it. Moreover, the geographical location of the study should be shown.
4. The three growth substances in lines 16-17 were not presented clearly.
5. In lines 16-19, this sentence should be rewritten.
6. The methods in the abstract lacked important details. What heavy metals did your study target? It should be listed in line 15. What was the source of the fungi? Where were they isolated from?
7. In the keywords, names of species should be italicized.
8. In lines 59-69, some specific species of AMF should be raised as examples for their functions.
9. Why was only P uptake focused on, instead of N uptake? Moreover, in the introduction, there was no argument stating the importance of P to plants and the reason why the current study concentrated on the P uptake.
10. Zinc and lead contamination should appear in the keywords.
11. Were there any literatures for the soil analysis?
12. The names of the black locust and maize were not italicized in lines 126 and 132, this should be corrected and checked throughout the paper.
13. In line 158, the format of HNO3 was in incorrected, this should be checked again for the whole paper.
14. The tables should not be filled with any colors.
15. The way to use the uppercase letters to indicate differences between substrates was unclear, the table 2 should be reformatted for a better readability.
16. Figure positions should be centered.
17. The discussion should refer the differences between the three substrates in Table 1 to explain the differences in their performance.
Minor editing of English language required
Author Response
On behalf of my co-authors, we would like to express our great appreciation to you for taking the time to review our paper and giving these comments to improve our manuscript. We have carefully considered your suggestion and made some changes. We also have tried our best to modifiy confusing sentences and entrusted the manuscript to a dissertation polishing company for improvement. We hope our revisions and responses will meet your expectations. Please see the attachment.
